# Neutron Star Binary Mergers: The Legacy of GW170817 and Future Prospects

Giulia Stratta [1,2,3,4,*,†] and Francesco Pannarale [4,5,†]

1 Institut für Theoretische Physik, Goethe Universität, Max-von-Laue-Str. 1, 60438 Frankfurt am Main, Germany
2 Istituto di Astrofisica e Planetologia Spaziali, via Fosso del Cavaliere 100, I-00133 Roma, Italy
3 Osservatorio di Astrofisica e Scienze dello Spazio, via Gobetti 93, I-40135 Bologna, Italy
4 INFN Sezione di Roma, Piazzale A. Moro 5, I-00185 Roma, Italy; francesco.pannarale@roma1.infn.it
5 Dipartimento di Fisica, Università di Roma "Sapienza", Piazzale A. Moro 5, I-00185 Roma, Italy
* Correspondence: giulia.stratta@inaf.it
† These authors contributed equally to this work.

**Abstract:** In 2015, the Advanced Laser Interferometer Gravitational-wave Observatory (LIGO) and Advanced Virgo began observing the Universe in a revolutionary way. Gravitational waves from cosmic sources were detected for the first time, confirming their existence predicted almost one century before, and also directly revealing the existence of black holes in binary systems and characterizing their properties. In 2017, a new revolution was achieved with the first observation of a binary neutron star merger, GW170817, and its associated electromagnetic emission. The combination of the information from gravitational-wave and electromagnetic radiation produced a wealth of results, still growing, spectacularly demonstrating the power of the newly born field of gravitational-wave Multi Messenger Astrophysics. We discuss the discovery of GW170817 in the context of the achievements it brought to Gamma-Ray Burst astrophysics, and we also provide a few examples of advancements in fundamental physics and cosmology. The detection rates of binary neutron star mergers expected in the next decade for third generation gravitational-wave interferometers will open the new perspective of a statistical approach to the study of these multi-messenger sources.

**Keywords:** gravitational wave; gamma-ray burst; GW170817

## 1. Introduction

During the last eight years, we witnessed a major revolution in physics, astrophysics and astronomy, one that will keep shedding profound implications in the decades ahead of us. On 14 September 2015, shortly after commencing the first observing run (O1), the two Advanced LIGO (Laser Interferometer Gravitational-wave Observatory) detectors [1] directly observed a gravitational-wave (GW) signal for the first time [2]. This signal, known as GW150914 from its discovery date, was shown to be emitted by the merger of two black holes (BHs) at a luminosity distance of ∼400 Mpc, with masses of about 36 and 29 solar masses. The observation of GW150914 constitutes a landmark in mankind's understanding of the universe: it initiated the era of GW astronomy and led to the 2017 Nobel Prize in Physics. GWs have remarkably small amplitudes, typically below 1 part in $10^{21}$ for the most violent events in the universe, but carry precious and otherwise inaccessible information about the emitting source. As an example, the first detection demonstrates that binary BHs exist and that they merge within a Hubble time.

Other two GW signals emitted by a pair of merging BHs were uncovered in O1, which ended on 19 January 2016, but it was the subsequent run (O2) that presented a second historical milestone. O2 lasted roughly twice as much as O1, from 30 November 2016 to 25 August 2017, and saw the participation of the Advanced Virgo detector [3] in observing operations. This enabled the first three-detector observations of GWs [4]. Eight

GW signals were observed during O2 [4]. Among them is GW170817 [5], the first GW detection compatible with the inspiral of a binary neutron star (BNS), a system formed by two neutron stars (NS) at the end of its evolution. The source of GW170817 was located at ~40 Mpc.

GW170817 was followed by GRB 170817A [6], a short gamma-ray burst (GRB) detected independently by the *Fermi* Gamma-ray Burst Monitor [7], and the Anti-Coincidence Shield for the Spectrometer for the *International Gamma-Ray Astrophysics Laboratory* (*INTEGRAL*) [8]. The initial LIGO-Virgo sky-localization of GW170817 had an area of about 30 $deg^2$ and an extensive follow-up observing campaign was launched throughout the electromagnetic spectrum [9]. Within ~11 h from the merger of the BNS source of GW170817, multiple teams independently detected a transient source in the optical/near-infrared bands, AT2017gfo, in NGC 4993, a galaxy at a distance compatible with the one obtained from the GW data analysis, ~40 Mpc, and contained within the 90% confidence sky-localization area of GW170817 (e.g., [10] and references therein). The transient source resulted to be consistent with a kilonova (e.g., [11,12]). Already predicted since the 1970s [13,14], the term "kilonova" indicates an emission component ~1000 times brighter than a nova. Kilonova radiation is powered by the radioactive decay of heavy unstable isotopes, the nucleosynthesis of which invokes rapid neutron capture processes allowed in core-collapse supernovae but also in the neutron-rich environment generated after the merger of two NSs. The observations that followed targeted AT2017gfo and its environment, unveiling emission across the whole electromagnetic spectrum. The totality of the GW and electromagnetic observations, including the untargetted detection of GRB 170817A, constitute the first multi-messenger observation with GWs (Figure 1).

The number of GW observations increased even more dramatically in the third observing run (O3), which covered two distinct stretches of time—1 April 2019 to 1 October 2019 and 1 November 2019 to 27 March 2020 [15–17]—and saw the participation of a fourth GW interferometer located in Japan, KAGRA [18] near the end of the run. As of today, the LIGO-Virgo-KAGRA Collaboration has announced a total of 90 GW observations and other searches on publicly available data have reported additional candidates [19–25]. However, no electromagnetic counterpart other than GRB170817A/AT2017gfo was found so far.

Accurate sky localization and/or source proximity are crucial parameters for a successful detection of the electromagnetic counterpart within the nearby distances explored by the current GW interferometer network for the detection of BNS or NS-BH mergers. This is because, due to geometrical reasons, in the local Universe we detect more often sources at large viewing angles (off-axis with respect to the jet main direction) rather than sources pointing their jet towards Earth (on-axis). At large viewing angles, the radiation is expected to be fainter and softer than that of bright gamma-ray bursts. The low luminosity of such off-axis radiation makes its identification challenging for sky surveys for sources at distances comparable to the maximum reach of second generation GW detectors. For instance, the isotropically radiating kilonova component is expected to challenge the follow-up campaigns for its identification already above few hundreds of Mpc. Other possible electromagnetic components less collimated and much fainter than the prompt emission from gamma-ray bursts are expected to come from the slower-expanding lateral sides of the relativistic jet launched soon after the merger, or from the expanding cocoon formed by the jet interaction with the surrounding matter ejecta released after the merger (e.g., [26] and references therein). Given the expected intrinsic faintness of off-axis radiating components, the more nearby and accurately localized the GW source is, the higher the electromagnetic counterpart detection chances are.

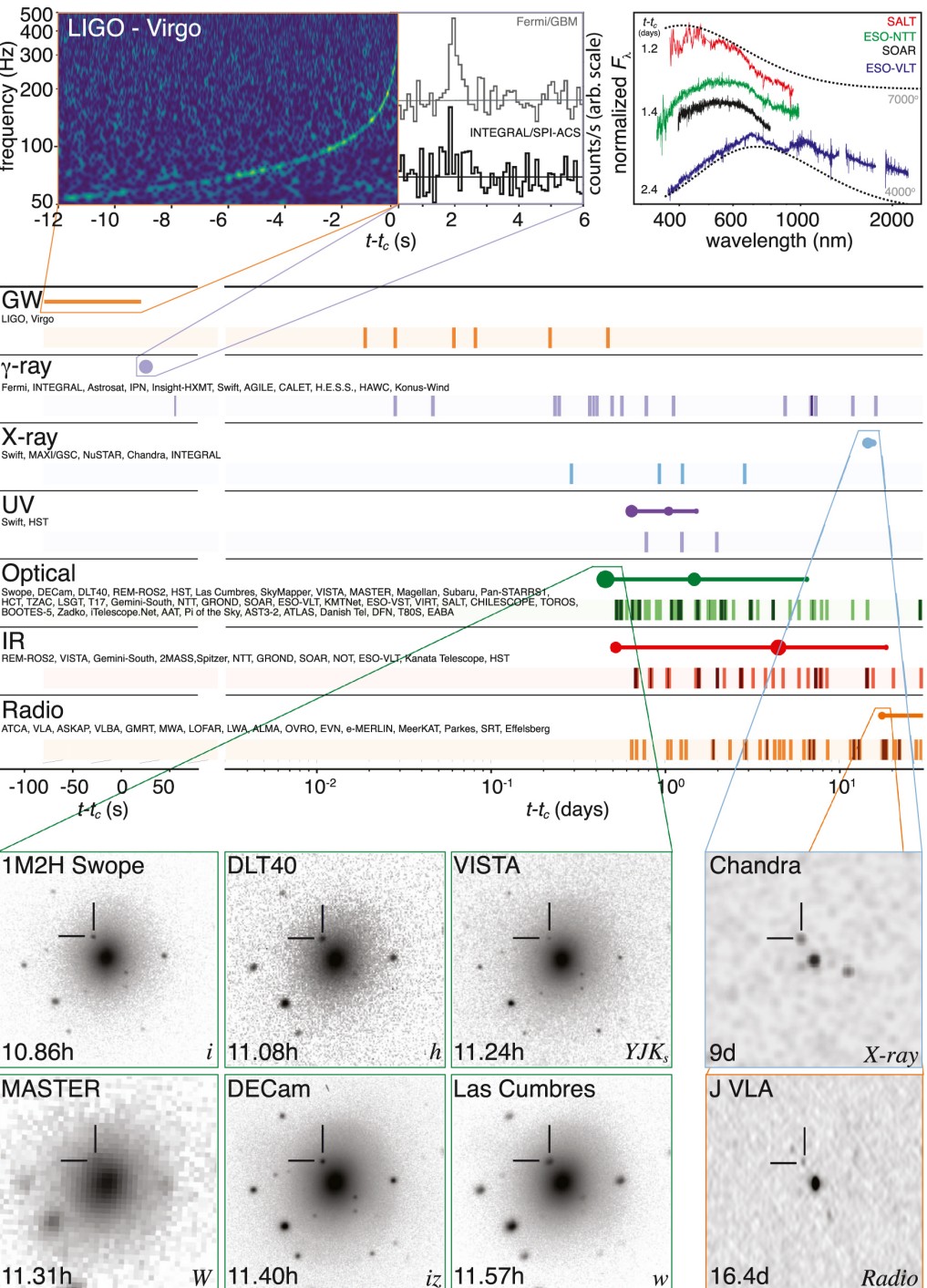

**Figure 1.** Top left panel: Gravitational wave signal of GW170817 represented in the time-frequency domain, and the associated short GRB detected independently with Fermi and INTEGRAL. Temporal scale is centered on the coalescence epoch $t - t_c = 0$ and clearly shows how the GRB trigger lagged 1.7 s the merger. Top right panel: Kilonova (AT2017gfo) nearly thermal optical spectrum taken at different epochs. Central panel: Timeline of the discovery of the electromagnetic counterparts (GRB 170817A and AT2017gfo) during the follow-up campaign. Bottom left panel: Images of AT2017gfo and the host galaxy NGC 4993. Bottom right panel: Images of the X-ray and radio afterglow of GRB 170817A. Figure taken from [9].

Among the signals uncovered by the LIGO-Virgo-KAGRA Collaboration, in the context of this article, we highlight GW190425 [27], a second confident detection compatible with a BNS source, and GW200105 and GW200115 [28], two confident detections compatible

with the binary merger of an NS and a BH. Indeed, mergers of compact objects containing at least one NS are the best candidates for emitting also electromagnetic radiation as well as neutrinos, and are therefore of greater interest for multi-messenger astrophysics with respect to binary BH mergers. In the case of GW190425, GW200105_162426 and GW200115_042309 (often abbreviated as GW200105 and GW200115), no electromagnetic or neutrino counterpart was observed, leaving GW170817 as the only GW multi-messenger event so far. Notably, however, GW170817 is the GW event with the highest signal-to-noise ratio (∼32) observed to date. Its initial localization area was much smaller than those of GW190425, GW200105, and GW200115 (10,200 deg$^2$, 7700 deg$^2$, and 900 deg$^2$, respectively) and the same is true for the median source luminosity distance (∼40 Mpc versus 159 Mpc, 280 Mpc, and 300 Mpc). These properties alone could explain the absence of any electromagnetic counterpart from these sources, though other explanations are still viable. Event GW190814 also deserves a special mention. Its source is a compact binary coalescence involving a 22.2–24.3 $M_\odot$ black hole and a compact object with a mass of 2.50–2.67 $M_\odot$ (90% credible intervals) the identification of which is still under debate (i.e., a massive NS or a light BH). The source is compatible with being at a distance of ∼240 Mpc [29]. The gravitational wave event was detected with signal-to-noise ratio of 25 by the two LIGO detectors and Virgo, allowing for the best sky localization ever achieved for a GW source, a 90% probability area of 18.5 deg$^2$. The detection of GW190814 promptly triggered a vast electromagnetic follow-up campaign. However, despite the excellent sky-localization, no electromagnetic counterpart was found for this source [30,31].

In this work, we provide a description of the multi-messenger observations and main achievements obtained from GW170817 and GRB 170817A accessible to the non-expert reader. For a general review on GRB physics and NS mergers, as well as their observational signatures, we recommend the reader to [32] and [33], respectively. Here, after a short introduction on GRBs and the long quest for the origin of short GRBs in Section 2, in Section 3 we provide an overview of the GW and electromagnetic radiation observations and a few examples of the implications achieved from their joint analysis, for the only case for which such analysis could be performed, that is the BNS associated with GW170817 and the short GRB 170817A. In Section 4 we describe the expectations from the observing runs of LIGO/Virgo/KAGRA that will be carried out in the forthcoming years, as well as from the observations that will be taken in the next decade when the next generation ground-based GW interferometers will be operational.

## 2. The Long Quest for the Short GRB Progenitor

Discovered during the Cold War epoch in the late 1960s by the military US satellites VELA, GRBs appear as serendipitous, intense flashes of gamma-rays from an unpredictable direction in the sky, and lasting a very short time (i.e., seconds, [34]). During the 1990s, thanks to the *Burst and Transient Burst Experiment* on board the *Compton Gamma-ray Observatory*, thousands of GRBs were observed, though no robust clues on their origin were achieved. By studying the burst duration distribution, however, a clear bimodal behaviour was found [35]. This result allowed for the first time to point out the existence of two classes of GRBs, likely originating from different progenitor systems. The empirical definition of "short" and "long" GRBs was introduced at that time to identify GRBs lasting less and more than ∼2 s, respectively, where long GRBs are also spectrally softer than the short ones (see [36] for a recent review).

By the end of the 1990s, the rapid slew capabilities of the X-ray telescopes onboard the *BeppoSAX* satellite (e.g., [37]) and, later on, of the *Swift* satellite [38], allowed for the discovery of an afterglow emission of GRBs. The study of the afterglow properties led to huge step forwards in our understanding of the nature of GRBs. Initially detected as soft X-ray non-thermal sources, afterglows have an intensity that quickly evolves towards lower frequencies down to the radio bands on timescales of the order of hours-days. Once the optical afterglow has faded away, the GRB host galaxy could be detected for several events at the position of the afterglow. Together with the study of redshifted spectral line systems,

the presence of a spatially coincident host galaxy directly confirmed the cosmological (i.e., non-galactic) origin of GRBs.

From optical afterglow long monitoring, almost all long GRBs observed so far at sufficiently nearby distances (i.e., at redshift $z < 1$) were found to be spatially and temporally consistent with supernova (SN) signals compatible with the core-collapse of massive stars (e.g., [39]). The associated supernovae are of type Ib and Ic, for which no hydrogen, and sometimes helium, lines are found, consistent with a Wolf-Rayet like progenitor star. The supernova signal typically rises above the afterglow flux level for a few to tens of days after the burst epoch in the optical bands, when the bright, non-thermal GRB afterglow emission has faded away. A spectacular evidence of a SN rise from the fading afterglow emission can be observed in the case of the long GRB 030329 [40].

Contrary to long GRBs, direct evidence of the nature of short GRB progenitors lacked for a long time and could not be obtained until the observation of GW170817/GRB 170817A. Nevertheless, a wealth of indirect evidence gathered in the past decades from short GRB afterglow and host-galaxy observations suggested a connection with a stellar population older than the one generating long GRBs (see, e.g., [41] and references therein). The following are some of the most prominent results in this sense:

- No core-collapse supernova is typically found to be associated with short GRBs, even for the most nearby ones: this strongly disfavours any origin linked to the gravitational collapse of massive stars.
- The position of short GRBs within the host galaxy typically tracks the outermost regions rather than central, bright, star-forming ones. Such a spatial distribution is more consistent with an old stellar population that had enough time to migrate far from its formation region, rather than with a young, short-lived one [42].
- A further clue in favour of an older progenitor is based on the mixed type of short GRB host galaxies, with about half of them being elliptical and the other half being forming galaxies, depending on the merger time delay [43].
- Finally, in some short GRBs, a rising thermal optical/near-infrared emission was detected and identified with kilonova emission.

These properties of short GRBs are consistent with a BNS merger progenitor, but also an NS-BH one (e.g., [44]). Fully unveiling the identity of the population of short GRB progenitors is the next great quest for future multi-messenger observational campaigns. This includes determining whether indeed NS-BH mergers can ignite short GRBs.

We remark that the identification of the GRB progenitor from the prompt emission duration alone is sometimes unreliable. For instance, the recent event GRB 211211A localized at 350 Mpc [45], i.e., one of the nearest GRB ever discovered, lasted up to ∼50 s and was therefore classified as a long one. At the same time, a kilonova component has been observed in its afterglow emission, strongly in favour of a BNS origin rather than a massive star collapse [45]. On the other hand, another recent example is GRB 200826A, that was classified as a short GRB, with a duration of ∼0.5 s, but for which a temporally and spatially consistent supernova signal was clearly detected, thus allowing to infer its association with the collapse of a massive star [46]. Therefore, as a general rule, a suite of observables should be taken into account to correctly identify the GRB progenitor. Yet, the most striking evidence in favour or against a compact binary merger progenitor is the detection or non detection of GWs, providing the source is within the sensitivity reach of GW interferometers.

## 3. GW170817/GRB 170817A: Observations and Implications

GW170817 [5] is the first GW source identified with the merger of a BNS system and the first GW source to be spatially and temporally associated with large statistical significance with an independently detected electromagnetic source. The identification of the source with a short GRB (GRB 170817A) cemented the association of this well known celestial phenomenon to the merger of compact objects, BNSs in particular.

This section describes the GW and electromagnetic signal properties of this source in Sections 3.1 and 3.2, respectively, while in Section 3.3 we show how the combination of these observations allowed us to tackle issues related to fundamental physics, GRB jet properties, and cosmology.

### 3.1. The Gravitational-Wave Signal

GW170817 was observed by Advanced LIGO and Advanced Virgo on 17 August 2017 at 12:41:04 UTC. 1.1 s prior to this time, which marks the coalescence time of the source of GW170817, data from the LIGO-Livingston detector presented a short instrumental noise transient which produced a $\lesssim$5 ms saturation of unknown cause in the digital-to-analog converter of the feedback signal controlling the position of the test masses [5]. The search analyses mitigated the effect of this noise transient by windowing to zero the data around it, while the source parameter estimation inference analyses also modelled the noise transient with a time-frequency wavelet reconstruction and subtracted it from the data. The initial analyses used data starting from 30 Hz, while the final, refined inference analyses were extended down to 23 Hz [47].

GW170817 was detected with a combined signal-to-noise ratio of 32.4—with single detector values of 18.8, 26.4 and 2.0 in LIGO-Handford, LIGO-Livingston and Virgo, respectively—and a false-alarm-rate estimate of less than one in $8.0 \times 10^4$ years. The source was rapidly localized to be within a 90% credible sky area of 190 $\deg^2$ using LIGO only data, and then to 31 $\deg^2$ when folding in Virgo data as well. In higher latency this value was brought down to 28 $\deg^2$, while the final analysis, based only upon GW data, further reduces it to 16 $\deg^2$. GW170817 remains the most precisely localized GW signal.

The physical parameters of the source were inferred using several gravitational waveform models that incorporate finite size effects, i.e., the effects on the waveform phase of the tidal field of one mass on its companion [47]. Two families of priors were used, differing in the maximum admitted value for the magnitude of the dimensionless rotational angular momentum parameter (spin) of each binary component.[1] The low-spin prior assumes isotropic and uncorrelated orientations for the spins, and a uniform prior for the magnitude up to 0.05, while the high-spin prior allows values up to 0.89. The 90% credible intervals for the primary and secondary mass of the source of GW170817 are $(1.36, 1.89)\, M_\odot$ and $(1.00, 1.36)\, M_\odot$, respectively, in the source frame and using the high-spin prior. These intervals reduce to $(1.36, 1.60)\, M_\odot$ and $(1.16, 1.36)\, M_\odot$ with the low-spin prior. The binary inclination angle $\iota$, which as we will see in the next section is of particular interest in the context of GRB physics, was measured to be $146^{+25}_{-27}$ deg with the low-spin prior and $152^{+21}_{-27}$ deg with the high-spin prior, where the subscript and superscript are the 5% lower limit and 95% upper limit, respectively, and the base number is the median. When folding into the prior the distance to NGC 4993, these measurements are $151^{+15}_{-11}$ deg and $153^{+15}_{-11}$ deg: the increased precision obtained in both cases highlights the power of multi-messenger observations, over single-messenger ones.

### 3.2. The Electromagnetic Signal

In the electromagnetic spectrum, the source appeared as a short GRB, GRB 170817A, that triggered the *Fermi*/GBM and *INTEGRAL*/ACS detectors $(1.74 \pm 0.05)$ s after the merger epoch (Figure 1) with locations compatible with that of GW170817 [6]. The observational properties of the burst were consistent with typical short GRBs observed by *Fermi*/GBM; however, its intrinsic energy and luminosity were several orders of magnitude below the values registered for standard short GRBs (e.g., [7,48]). The significance of the coincident (in time and space) detection of GRB 170817A and GW170817 was calculated using the 28 $\deg^2$ uncertainty GW skymap and the *Fermi*/GBM skymap with an area of 1100 $\deg^2$. The odds of the common-source hypothesis to the unrelated signal source hypothesis were found to be $\gtrsim 10^6$, a decisive evidence for the common origin of the GW and GRB events [49].

No bright emission at longer wavelengths was immediately detected by large field of view telescopes, within the initial sky-localization area of GW170817. However, by monitoring the galaxies contained within the sky-localization area at known distance compatible with the one obtained from the GW data analysis, an optical transient, AT2017gfo, was clearly detected with narrow field and more sensitive telescopes [50]. The optical counterpart detection happened about 11 h after the merger epoch (Figure 1). The source faded on a time scale of a few weeks and showed a clear reddening trend. The intrinsic luminosity, temporal evolution and spectral properties of AT2017gfo were consistent with those expected from a kilonova, and the consistency of AT2017gfo with the expected kilonova properties allowed for it to be associate with GW170817.

In contrast with the initially bright and fast fading afterglow emission of a typical GRB, the afterglow of GRB 170817A was detected for the first time with the *Chandra* X-ray telescope only 9 days after the merger epoch as a faint, slowly rising X-ray source, positionally consistent with AT2017gfo [51]. The afterglow was detected with similar temporal properties also in radio and optical bands. At about 150 days after the burst epoch, the afterglow emission reached its maximum value at all wavelenghts, and started to decay following a power law. Today (mid 2022), the source is still under scrutiny by various observatories and very recent results from deep exposures revealed that the power-law decay is continuing and fluxes are reaching the detection thresholds both in X-rays [52] and radio [53].

The low energy content of GRB 170817A and the peculiar temporal behaviour of its afterglow resulted to be consistent with prediction from a lateral view of the system, compatible with a viewing angle range with respect to the jet axis of the order of 10–30 deg where the viewing angle is defined as $\min(\iota, 180° - \iota)$ in terms of the binary inclination angle used in GW physics.

### 3.3. Implications of Multi-Messenger Data Analysis Results

In this section, we provide a few examples of how GWs and electromagnetic radiation from the same source can be combined, thus illustrating the power of multi-messenger data analysis in several fields ranging from fundamental physics to cosmology.

### 3.3.1. Propagation Speed of Gravitational Waves

The time delay between the burst epoch of GRB 170817A and the merger epoch of GW170817 constrains the GW propagation speed $v_{GW}$ with respect to the speed of light $v_{EM}$ [6]. By assuming that the GW travel time is $t_{GW} = t_{EM} \pm \epsilon$, then for small $\epsilon$ we can write the difference in propagation speed between photons and GWs, $\Delta v = v_{GW} - v_{EM}$, as $D|\epsilon|/(t_{EM}^2 \pm \epsilon t_{EM}) \sim D|\epsilon|/t_{EM}^2$ and the fractional speed difference as $\Delta v/v_{EM} \sim v_{EM}(|\epsilon|/D)$, where $D$ is the travel distance.

Following Abbott et al. [6], by assuming that the two messengers were emitted simultaneously, the observed time delay of 1.7 s in the arrival time of the gamma-rays is only due to the higher propagation speed of GWs with respect to photons. In this case, $\Delta v/v_{EM} = v_{EM}(\epsilon/D) < 7 \times 10^{-16}$, where we substituted $v_{EM}$ with the speed of light, $\epsilon$ with 1.7 s, and $D$ with the $1\sigma$ lower bound on the source luminosity distance, i.e., 26 Mpc ($7.8 \times 10^{25}$ cm), to obtain a conservative upper bound on $\Delta v/v_{EM}$. Alternatively, the measured time difference may not represent $\epsilon$ if an intrinsic delay $\tau$ between the GW peak emission and the photon emission is present, for instance due to the time required for the jet to reach the photosphere while carving the ejecta released after the merger. In this case, the measured time difference of 1.7 s corresponds to $\epsilon + \tau$. By assuming $\tau \lesssim 10$ s, then $\epsilon \gtrsim -8.3$ s implies a lower propagation speed of GWs with respect to photons, with $\Delta v/v_{EM} = v_{EM}(\epsilon/D) \gtrsim -3 \times 10^{-15}$.

As stated in Abbott et al. [6], while the emission times of the two messengers are model dependent, simple conservative assumptions led to significant improvements over previously existing constraints.

### 3.3.2. Short GRB Jet Structure

The equivalent isotropic energy $E_{iso}$ of GRB 170817A was found to be several orders of magnitude lower than the typical values observed for short GRBs [6]. The afterglow of GRB 170817A was also anomalous, showing an initial slow rising behaviour never observed previously in short GRBs. These observational properties resulted to be consistent with a large viewing angle of the observer with respect to the jet axis. Such a lateral view of the jet and the proximity of the source allowed to investigate the jet structure of short GRBs for the first time.

In order to explain the GRB 170817A energy content and afterglow behaviour, a simple "Top Hat" geometrical jet model, where the energy content and expansion velocity are uniform within the jet cone, requires an ad hoc position of the observer, with viewing angle at the very edge of the jet cone. For this reason, this jet model has been considered as less favourable. A more realistic scenario is represented by a "structured" jet model, where the energy and velocity distributions are uniform within a very narrow central core with half opening angle $\theta_c$, while for $\theta > \theta_c$ they scale with $\theta$, thus explaining the low energy content of GRB 170817 from an "off-axis" observer. Different models assume a power-law or gaussian function for the angular dependency, both of which can describe the observed properties.

An alternative interpretation invoked the presence of a poorly collimated ("cocoon") and mildly relativistic ejecta formed by the interaction of the jet with the surrounding dense material released after the merger process [54]. To disentangle the two scenarios (structured jet vs. cocoon), radio observations resulted to be crucial. High-spatial resolution measurements of the source size and displacement were obtained with the Very Long Baseline Interferometry observations months after the merger. These observations showed a clear displacement of the radio source, consistent with a non-expanding source size, thus indicating the presence of a narrow, relativistic jet and at odds with an expanding, nearly isotropic cocoon structure [55,56].

Over the recent years, Particle-in-cell (PIC) simulations have gained particular relevance in studying jet structure and radiative properties, given the increasing computing power available in high-performance computing facilities. These simulations represent an extremely powerful tool for multi-messenger astrophysics since they allow to investigate how GRB radiation is generated from relativistic jets in BNS and NS-BH systems (see, e.g., [57] for a recent review).

### 3.3.3. Cosmology

Coalescing binaries of NSs and BHs are known as standard sirens, as they allow for a direct measurement of the luminosity distance to the source $D_L$ via GWs, the amplitude of which scales with the inverse of this distance. On the other hand, GWs cannot provide a measurement of the source redshift $z$: because they are invariant under the redshift transformation that rescales masses, lengths, and frequencies, GWs are insensitive to the value of $z$. An independent measurement of $z$, however, provides a probe of cosmology when combined with the GW measurement of $D_L$ [58–61], which is of utmost importance given the current tension between different probes.

The recession velocity $v_H$ of the source can be inferred from $z$, and thus the Hubble constant $H_0$ can be determined using the joint multi-messenger data from the well known formula $v_H = H_0 D_L$, which is applicable in the local Universe.

The multi-messenger observation kickstarted by GW170817 yielded a cosmological measurement of this kind. An analysis of the GW data on its own provided a source distance value of $\sim$40 Mpc; once the host galaxy of the event, NGC 4993, was identified, the Hubble velocity of the source was found to be $\sim$3000 km/s.

The combined data for the two messengers thus gave a Hubble constant value of about about 70 km per second per megaparsec [62]. The maximum posterior value sits between the Plank and SHoES $H_0$ results, but has much larger uncertainties, and thus cannot alleviate the tension between the two results. However, repeated and combined measurements of

this kind will reduce the uncertainty on the multi-messenger measurement of $H_0$ and thus provide an extremely relevant contribution to cosmology.

### 3.3.4. Heavy Element Formation

The temporal and spatial association of the optical transient AT217gfo with the GW event GW170817 (see Section 3.2) represents the first robust evidence that, after the merger of two NSs, an optical/near-infrared thermal emission is developed on timescales of the order of days/weeks (Figure 1). The luminosity, temporal and spectral properties of AT2017gfo are consistent with a "kilonova" (e.g., [11,12,63,64]), i.e., quasi-thermal emission from the ejected material, and thought to be powered by the radioactive decay of freshly synthesized elements heavier than Iron (e.g., [64] and reference therein). As a consequence, the association of AT2017gfo with GW170817 witnesses the potentially crucial role of BNSs in the nucleosynthesis of those elements that are not produced through thermonuclear reactions inside stars.

In particular, the degradation of the high-energy photons produced by the radioactive decay of freshly formed elements into the optical/near-infrared bands stems from the large opacity produced by the scattering with complex outer electron shells, as for the case of the lanthanides (atomic number $Z = 57$–71). These elements require extreme temperatures and free-neutron densities to trigger the so-called "rapid neutron capture" nucleosynthesis, where the $\beta^-$ decay timescale, within which a neutron decays into a proton (with the emission of an electron and an antineutrino), is longer than the timescales for the nucleus to capture a neutron from the external environment. The *r*-process nucleosynthesis (where *r* stands for rapid) is at the origin of half of the elements heavier than Iron and BNSs, along with core-collapse supernoavae, represent the major cosmic sites for their production.

Marginal evidence of kilonova emission had already been observed in the past against a few short GRB as an additional optical/near-infrared component emerging from the afterglow light curves a few days after the burst trigger (e.g., [65]), but at distances $z > 0.1$ such that the signal was too faint to obtain any detailed spectral or temporal analysis. Due to its proximity to the Earth, the high-quality data obtained for AT2017gfo allowed us to investigate deeply the spectral properties of a kilonova for the first time. On one hand, the thermal nature was robustly confirmed; on the other, however, the strong line blending due to Doppler broadening of the mildly relativistic expanding ejecta ($\sim 0.2 - 0.3c$) prevented any unambiguous identification of elements [11,12], though recent studies from synthetic spectra comparison show the presence of Strontium [66].

## 4. Future Perspectives

Expected GW and electromagnetic joint detection rates for BNS systems are still very uncertain. This is not surprising as so far we have detected simultaneous GW and electromagnetic radiation only from one source (GW170817/GRB 170817A). Furthermore, obtaining these predictions requires folding together several uncertain variables and aspects, as for instance the true BNS merger rate, the sensitivity and duty cycle of GW detectors in future runs, the GW observation strategy (for example, all-sky searches vs. searches that target a specific GRB event), the observation strategy and performance of electromagnetic facilities, and models to connect the single BNS progenitor to its GRB prompt and afterglow emission, as well as to its kilonova emission.

Current estimates of the BNS merger rate density for different population models are reported in Abbott et al. [67]. Taking the lowest 5% and highest 95% credible interval of these models, the merger rate is found to be between $10\,\mathrm{Gpc}^{-3}\,\mathrm{yr}^{-1}$ and $1700\,\mathrm{Gpc}^{-3}\,\mathrm{yr}^{-1}$. When assuming that the masses of NSs in merging binaries are uniformly distributed between 1 and 2.5 solar masses, that the merger rate is constant in comoving volume up to $z = 0.15$, and that the individual NS spin magnitudes are distributed uniformly between 0 and 0.4, the BNS merger rate is found to be $105.5^{+190.2}_{-83.9}\,\mathrm{Gpc}^{-3}\,\mathrm{yr}^{-1}$, to be compared to the $320^{+490}_{-240}\,\mathrm{Gpc}^{-3}\,\mathrm{yr}^{-1}$ found in Abbott et al. [27] under the same assumptions.



Second generation detectors, i.e., LIGO, Virgo and KAGRA, are nominally expected to be able to detect BNS systems up to 190 Mpc, 115 Mpc and 10 Mpc, respectively, by the end of the next observing run (O4), and 325 Mpc, 260 Mpc and 128 Mpc, respectively, by the end of the following one (O5).[2] The typical bright prompt emission of short GRBs requires small viewing angles in order to be detected. Therefore, to achieve a detection rate > 1 per year, large volumes of Universe are required, that is, distance reaches much larger than the one achieved by the second generation detectors. The GRB faint prompt emission as for GRB 170817A and afterglow are likelier to be observable as they do not require a viewing angle as optimal as that of the bright prompt emission. Finally, kilonova emission is isotropic and can be detected with any orientation of the BNS system; thus, a joint detection of the GW radiation with this kind of electromagnetic counterpart is the likeliest.

Overall, the estimated GW+GRB joint detection rate values for the next years (up to the first half of the 2030s) are compatible with a lower limit of less than one per year during the next GW observing run when considering the bright prompt GRB emission (see [68–70], and references therein); however, this rate can be driven up to ∼6 per year when considering specific BNS population scenarios coupled with *Fermi*/GBM's large field of view [69]. We note that Colombo et al. [70] do not find such high values, but they do highlight that the best approach to obtain a joint detection in this context is to rely on subthreshold GW searches that follow up GRB events. As mentioned previously, kilonova and GW joint detections yield intrinsically more promising rates. These could be of the order of up to 10 per year in the next GW observing run, with a lower limit of ∼1 (see Colombo et al. [70] and estimates in Patricelli et al. [69]). The values we quoted so far are for BNS sources, but possibly also coalescing binaries containing an NS and a BH are short GRB progenitors and some of them are expected to produce a kilonova emission not too dissimilar from the one of BNS mergers, but with some differences, e.g., [71]. The rates are even more uncertain for these sources, particularly in light of our lack of electromagnetic counterparts for the GW candidate events observed so far. Andreoni et al. [72] carried out a study in the context of target-of-opportunity electromagnetic observations with the Vera C. Rubin Observatory finding that ∼15 NS-BH mergers could be probed in search for a kilonova counterpart, with this number being mainly constrained by the duration of GW observing runs during Rubin operations (see also [73]).

By the second half of the 2030s, a revolution will come with the next generation of GW detectors, such as the Einstein Telescope [74] and Cosmic Explorer [75]. With one order of magnitude higher sensitivity, these interferometers will easily reach distances corresponding to the peak of star formation ($z{\sim}2$), with an expected compact binary coalescence detection rate of the order of $10^5$ per year. A large sample of joint GW+GRB detections is thus expected, allowing for the first time for a statistical approach to multi-messenger astrophysics that will uniquely address a number of open questions. Just to mention a few examples: *(i)* The observed delay time between the merger epoch and the first gamma-ray emitted photon $\epsilon + \tau$ (see Section 3.3.1) is a relevant parameter to investigate on the still poorly known GRB jet formation mechanism. With a statistical approach, it will be possible to disentangle $\tau$ from the relative propagation time $\epsilon$, as only the latter is expected to depend on distance (see Section 3.3, [6]). *(ii)* The detection/non-detection of GRBs from a large number of face-on BNS, and possibly from NS-BH binary systems, will allow us to probe the jet production efficiency in such systems, where the inclination angle information can be obtained from GW data analysis. *(iii)* With a large number ($N$) of $H_0$ measurements, combining the luminosity distance $D_L$ with the redshift from GRBs will decrease the uncertainty on the measurement by a factor $1/\sqrt{N}$, and will allow for $\lesssim$1% accuracy, a mandatory level of precision to disentangle the current tension among different $H_0$ measurements obtained from distinct probes as the Cosmic Microwave Background and type-Ia supernovae (e.g., [76]).

We note that to reach all these goals it is of utmost importance to guarantee the presence of a GRB-dedicated space mission during the second half of the 2030s capable not only of detecting the gamma-ray counterpart, but also of providing the sky-localization

accuracy required to permit multi-band follow-up from optical to radio, as for instance the project mission THESEUS (Transient High-Energy Sky and Early Universe Surveyor, e.g., [77,78]) that has been proposed to the last call for medium-size mission opportunity in the European Space Agency's science programme for a launch in 2037.

**Author Contributions:** G.S. and F.P. contributed equally to this work. All authors have read and agreed to the published version of the manuscript.

**Funding:** This research received no external funding

**Institutional Review Board Statement:** Not applicable.

**Informed Consent Statement:** Not applicable.

**Acknowledgments:** We thank the anonymous referees for useful comments and suggestions. G.S. acknowledges the support by the State of Hesse within the Research Cluster ELEMENTS (Project ID 500/10.006).

**Conflicts of Interest:** The authors declare no conflict of interest.

## Notes

1    This is given by $cS_i/Gm_i^2$, where $S_i$ and $m_i$ are the NS rotational angular momentum and mass, respectively, and $c$ and $G$ are the speed of light and the gravitational constant, respectively.

2    https://observing.docs.ligo.org/plan/ (accessed date 15 June 2022).

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
