# Peer review of "Neutron Star Binary Mergers: The Legacy of GW170817 and Future Prospects"

_universe, doi:10.3390/universe8090459_

Round 1

Reviewer 1 Report

The authors present an comprehensive analysis of the current status of research in the field of NS star mergers. The paper is well-written, but some changes are necessary before it can be considered ready for publication.

My comments are listed below:

- lines 38-53: I believe that introducing here the time lag is odd, without further motivation, which is  done better in Sec. 3.2.

- lines 64-90: I strongly recommend the authors to report about the search for the electromagnetic counterpart of GW 190814, which is believed to be a possible NS-BH merger. The search for its electromagnetic counterpart is the best example of what can be achieved today using both all-sky searches and  searches that target a specific GRB event. This is true in terms of both the best astronomical and the most important collaborations involved. Please have a look to:

https://ui.adsabs.harvard.edu/abs/2020MNRAS.499.3868T/abstract
https://ui.adsabs.harvard.edu/abs/2020A%26A...643A.113A/abstract

- lines 81-84: please rephrase, as this sentence is not clear. Maybe the authors are trying to say that the most promising (for what?) counterparts are outside the jet. This is confusing since the jet is the source of an electromagnetic counterpart, taht is the GRB. If the authors instead are trying to say that the most promising counterpart is not the GRB because it is collimated, this is not clear.

-line 150: forming -> star-forming
-line 153: "These properties ... " This sentence should be motivated more and/or supported with a reference. For example, the authors can consider the works from the groups of Mapelli or Belczynski.

- line 261: If epsilon is the modulus of t_EM-t_GW, there is no real need to introduce Delta t.

- line 381: which instruments? Please rephrase.

- line 416: this is true for most events, which will be too far and thus faint for the  detection of the off-axis prompt and afterglow emission. However, a minority of cases will be close enough to be detected even if off-axis.

Author Response

We thank the Referee for useful comments and suggestions that we implemented in the text accordingly. Please find below our answers to Referee's comments. 

----

The authors present an comprehensive analysis of the current status of research in the field of NS star mergers. The paper is well-written, but some changes are necessary before it can be considered ready for publication.

My comments are listed below:

- lines 38-53: I believe that introducing here the time lag is odd, without further motivation, which is  done better in Sec. 3.2.

AUTHORS: This information is now only in Sec. 3.2

- lines 64-90: I strongly recommend the authors to report about the search for the electromagnetic counterpart of GW 190814, which is believed to be a possible NS-BH merger. The search for its electromagnetic counterpart is the best example of what can be achieved today using both all-sky searches and  searches that target a specific GRB event. This is true in terms of both the best astronomical and the most important collaborations involved. Please have a look to:

https://ui.adsabs.harvard.edu/abs/2020MNRAS.499.3868T/abstract
https://ui.adsabs.harvard.edu/abs/2020A%26A...643A.113A/abstract

AUTHORS: we added a new paragraph dedicated to this event, citing the suggested references.

- lines 81-84: please rephrase, as this sentence is not clear. Maybe the authors are trying to say that the most promising (for what?) counterparts are outside the jet. This is confusing since the jet is the source of an electromagnetic counterpart, taht is the GRB. If the authors instead are trying to say that the most promising counterpart is not the GRB because it is collimated, this is not clear.

AUTHORS: we rephrased this part.

-line 150: forming -> star-forming

AUTHORS: done

-line 153: "These properties ... " This sentence should be motivated more and/or supported with a reference. For example, the authors can consider the works from the groups of Mapelli or Belczynski.

AUTHORS: we added a reference to the work by Pannarale et al. 2014 on the "Prospects for Joint Gravitational-wave and Electromagnetic Observations of Neutron-star-Black-hole Coalescing Binaries"

- line 261: If epsilon is the modulus of t_EM-t_GW, there is no real need to introduce Delta t.

AUTHORS: The referee is right, we removed Delta_t

- line 381: which instruments? Please rephrase.

AUTHORS: done

- line 416: this is true for most events, which will be too far and thus faint for the  detection of the off-axis prompt and afterglow emission. However, a minority of cases will be close enough to be detected even if off-axis.

AUTHORS: point ii) at line 416 we highlight that, through the large detection number of on-axis/face-on binary systems as GRBs, we will be able to estimate the jet production efficiency. Though also nearby off-axis systems can be detected, these will be only a few and not enough to build a statistically significant sample.  So we left this sentence unchanged.

Reviewer 2 Report

Review of Neutron Star Binary Mergers: the Legacy of GW170817 and Future Prospects

         by Giulia Stratta, Francesco Pannarale

This review is well organized for layman with scientific terms. It is very informative, and the importance of multi-message astrophysics is compactly described.

I would like to suggest authors to add a figure and a few references.

Even this review is for layman, I would like to suggest you to include one of figures suggested below.

Add a basic plot from Fig. 2 in Abbott, B.P.; et al. Multi-messenger Observations of a Binary Neutron Star Merger. Astrophys. J. Lett. 2017, 848, L12, 451

[arXiv:astro-ph.HE/1710.05833]. doi:10.3847/2041-8213/aa91c9

Fig. 1 or Fig. 11 in

Burns, E., Neutron Star Mergers and How to Study Them,

Living Reviews in Relativity, 23, 1, 4, 2020, 10.1007/s41114-020-00028-7, arXiv:1909.06085 

will be very useful for readers.

I think it is very important to show the basic idea of multimessage observations, possibly simplified version only with GW and gamma-ray with observational data.

I also suggest citing a new book of GRB:

Zhang, B. (2018). The Physics of Gamma-Ray Bursts. Cambridge: Cambridge University Press. doi:10.1017/9781139226530

With

Burns, E., Neutron Star Mergers and How to Study Them,

Living Reviews in Relativity, 23, 1, 4, 2020, 10.1007/s41114-020-00028-7, arXiv:1909.06085

For section 3.3.2 Short GRB jet structure:

Particle-in-cell (PIC) simulations are importance to investigate how GRB radiation is generated from relativistic jets.

For example,

Nishikawa, K.-I., Dutan, I, Köhn, C., Mizuno, Y., PIC methods in astrophysics: simulations of relativistic jets and kinetic physics in astrophysical systems,

Living Reviews in Computational Astrophysics (2021) 7:1 https://doi.org/10.1007/s41115-021-00012-0

In this review PIC simulations of relativistic jets are described including future systematic simulations of GRB radiation (5.8 Mergers of neutron stars and black holes and associated jets and 5.9 Future PIC simulations of electromagnetic radiation from relativistic jets generated by binary mergers).

Author Response

We thank the Referee for useful comments and suggestions that we implemented in the text accordingly. Please find below our answers to Referee's comments. 

-------
Review of Neutron Star Binary Mergers: the Legacy of GW170817 and Future Prospects

         by Giulia Stratta, Francesco Pannarale

This review is well organized for layman with scientific terms. It is very informative, and the importance of multi-message astrophysics is compactly described.

I would like to suggest authors to add a figure and a few references.
Even this review is for layman, I would like to suggest you to include one of figures suggested below.
Add a basic plot from Fig. 2 in Abbott, B.P.; et al. Multi-messenger Observations of a Binary Neutron Star Merger. Astrophys. J. Lett. 2017, 848, L12, 451
[arXiv:astro-ph.HE/1710.05833]. doi:10.3847/2041-8213/aa91c9
Fig. 1 or Fig. 11 in
Burns, E., Neutron Star Mergers and How to Study Them,
Living Reviews in Relativity, 23, 1, 4, 2020, 10.1007/s41114-020-00028-7, arXiv:1909.06085 
will be very useful for readers.
I think it is very important to show the basic idea of multimessage observations, possibly simplified version only with GW and gamma-ray with observational data.

AUTHORS: We follow the Referee's suggestion and added a representative Figure showing the power as well as the challenge of multi-messenger observations.

I also suggest citing a new book of GRB:
Zhang, B. (2018). The Physics of Gamma-Ray Bursts. Cambridge: Cambridge University Press. doi:10.1017/9781139226530
With
Burns, E., Neutron Star Mergers and How to Study Them,
Living Reviews in Relativity, 23, 1, 4, 2020, 10.1007/s41114-020-00028-7, arXiv:1909.06085

AUTHORS: We thank the Referee for this useful suggestion and we add both Burns et al. 2020 and Zhang 2018 references in the text

For section 3.3.2 Short GRB jet structure:
Particle-in-cell (PIC) simulations are importance to investigate how GRB radiation is generated from relativistic jets.
For example,
Nishikawa, K.-I., Dutan, I, Köhn, C., Mizuno, Y., PIC methods in astrophysics: simulations of relativistic jets and kinetic physics in astrophysical systems,
Living Reviews in Computational Astrophysics (2021) 7:1 https://doi.org/10.1007/s41115-021-00012-0
In this review PIC simulations of relativistic jets are described including future systematic simulations of GRB radiation (5.8 Mergers of neutron stars and black holes and associated jets and 5.9 Future PIC simulations of electromagnetic radiation from relativistic jets generated by binary mergers).

AUTHORS: We thank again the Referee for this additional important suggestion and we added a short paragraph at the end of Section 3.3.2 on PIC simulation role in our understanding jet radiative properties and GRBs from BNS and NS-BH.

Reviewer 3 Report

Dear Editor

The authors provide a brief review concerning  the discovery of the GW170817 event in the context of  the achievements in brought to Gamma-Ray Burst Astrophysics. Moreover,  the authors discuss the new perspectives related to this event. 

Some more specific comments: The paper is well written and contains results that will be useful to the scientific community. The presentation is very clear and instructive.  Where necessary the appropriate references were given. I suggest publication in  the present form.

Author Response

We thank the Referee for the positive comment. We provided small text changes in order to account for other Referee's comments. 

----
Dear Editor

The authors provide a brief review concerning  the discovery of the GW170817 event in the context of  the achievements in brought to Gamma-Ray Burst Astrophysics. Moreover,  the authors discuss the new perspectives related to this event. 

Some more specific comments: The paper is well written and contains results that will be useful to the scientific community. The presentation is very clear and instructive.  Where necessary the appropriate references were given. I suggest publication in  the present form.